# UniMoD: Efficient Unified Multimodal Transformers with Mixture-of-Depths

## Abstract

Unified multimodal transformers, which handle both generation and understanding tasks within a shared parameter space, have received increasing attention in recent research. Although various unified transformers have been proposed, training these models is costly due to redundant tokens and heavy attention computation. In the past, studies on large language models have demonstrated that token pruning methods, such as Mixture of Depths (MoD), can significantly improve computational efficiency. MoD employs a router to select the most important ones for processing within a transformer layer. However, directly applying MoD-based token pruning to unified transformers will result in suboptimal performance because different tasks exhibit varying levels of token redundancy. In our work, we analyze the unified transformers by (1) examining attention weight patterns, (2) evaluating the layer importance and token redundancy, and (3) analyzing task interactions. Our findings reveal that token redundancy is primarily influenced by different tasks and layers. Building on these findings, we introduce UniMoD, a **task-aware token pruning** method that employs a separate router for each task to determine which tokens should be pruned. We apply our method to Show-o and Emu3, reducing training FLOPs by approximately 15% in Show-o and 40% in Emu3, while maintaining or improving performance on several benchmarks.

## 1 Introduction

Unified multimodal transformers have received growing attention due to their ability to handle both generation and understanding tasks within a shared parameter space. Recent studies (Xie et al., 2024; Team, 2024; Ge et al., 2024b; Wang et al., 2024; Sun et al., 2023; Ye et al., 2024; Ma et al., 2024; Zhou et al., 2024; Liu et al., 2024a) have explored different approaches for unified transformers. These models can be broadly categorized into two types: one where both generation and understanding tasks are handled using a fully autoregressive method (Ge et al., 2024b; Team, 2024; Wang et al., 2024), and another where generation tasks are addressed using diffusion or flow matching techniques, while autoregressive methods are used for understanding tasks (Xie et al., 2024; Zhou et al., 2024; Ma et al., 2024). However, regardless of the approach, training these unified transformers remains time and memory intensive, primarily due to token redundancy and the computationally heavy attention mechanisms. In the past, few studies have explored efficient training strategies for these models. Therefore, developing efficient methods for training unified transformers remains a significant challenge.

The Mixture of Depths (MoD) approach has been employed in previous studies on large language models (LLMs)(Raposo et al., 2024) and multimodal large language models (MLLMs)(Luo et al., 2024; Wu et al., 2024b; Zhang et al., 2024) to prune tokens. MoD employs a router to assign weights to tokens based on their significance, enabling the model to selectively prune less important tokens and reduce computational overhead. However, applying MoD to unified transformers presents unique challenges that are worth exploring. A straightforward application of MoD (Lin et al., 2024b) to unified transformers involves pruning tokens from sequences of any task, selecting and removing a fixed proportion of tokens at interleaved layer.

However, as shown in Fig. 1(a), this straightforward approach leads to suboptimal performance. A single router struggles to retain all important tokens and eliminate all redundant ones of all tasks. This limitation arises because different tasks exhibit varying levels of token redundancy at different layers.

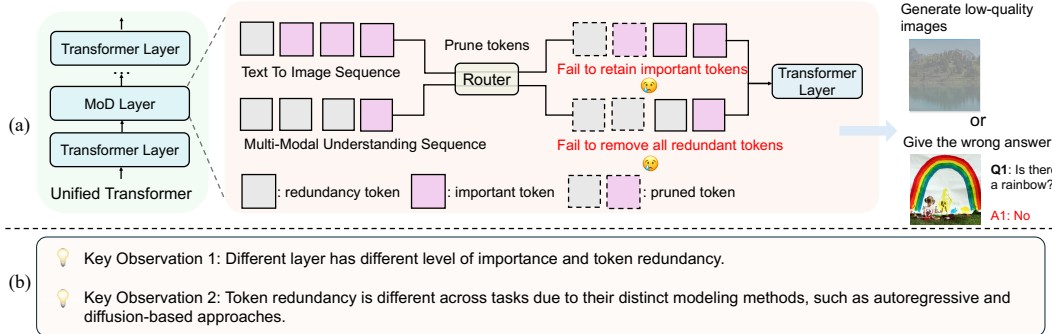

Figure 1: (a) Pipeline and challenges of applying Mixture of Depths (MoD) to unified transformers. A single router prunes tokens across tasks and layers, leading to suboptimal performance due to inconsistent token redundancy. (b) Two key observations from our experiments on unified transformers, providing critical insights for our proposed method.

Consequently, employing a single router to uniformly prune tokens for both tasks is ineffective, as it cannot accommodate the distinct redundancy patterns inherent to each task. Therefore, without a thorough analysis of these factors, developing a token pruning strategy that meets the diverse requirements of unified transformer models remains challenging.

In this work, we conduct an empirical analysis of unified transformers from three different perspectives and present an effective solution. (1) We analyze attention weights in several unified transformers to explore whether different tasks and modalities influence attention weight patterns. (2) We evaluate layer importance and token redundancy to determine which layers should be pruned through a series of simple experiments. Firstly, our inference experiments demonstrate that different layers exhibit varying levels of importance. We then use the ARank metric (Luo et al., 2024), which refers to Attention Map Rank, to analyze the differences in token redundancy across layers and tasks. Higher values of ARank indicate lower token redundancy. (3) We explore the interactions between different tasks through two experiments. Firstly, we conduct experiments to assess how removing one task affects the benchmark performance of the other task. Then, we introduce a competitive setting where tokens from different tasks compete for selection, allowing us to compare their relative importance.

Based on our experimental results and analysis, we draw two important observations as shown in Fig. 1(b). (1) Different layers within unified transformers exhibit varying levels of importance, with sequence importance and token redundancy fluctuating across layers. (2) The token redundancy is different across different tasks due to distinct modeling methods. For example, in the Show-o model, generation tasks use diffusion-based approaches, while understanding tasks rely on autoregressive methods. These differences lead to varying levels of redundancy and importance in tokens of the same modality (e.g., image or text). Therefore, from these observations, we propose UniMoD, a task-aware token pruning method for unified transformers. We transform several transformer layers into MoD blocks specialized for the generation task or the understanding task. Each task has its own router that assigns token weights, allowing the pruning process to adapt to each task.

In our experiments, we select Show-o and Emu3 as representative unified transformers. Show-o handles generation and understanding tasks with distinct modeling approaches, while Emu3 employs a fully autoregressive approach for both tasks. Our method is general and applicable to various unified transformer architectures, regardless of how tasks are modeled. It reduces FLOPs by 15% for Show-o and 40% for Emu3, while maintaining or improving performance on certain benchmarks. Additionally, our approach extends to pure diffusion-based generation models like PixArt (Chen et al., 2024b) and DiT (Peebles & Xie, 2023), demonstrating the versatility and integration capability of our method across diverse generation frameworks.

Our main contributions are as follows.

- We conduct an empirical analysis of examining attention weights, layer importance and token redundancy, and the interactions between different tasks in the unified transformers.

- We identify that different modeling methods for different tasks lead to significant differences in token redundancy and importance across tasks. Furthermore, within the same task, different layers exhibit substantial variations in token redundancy and importance.

- To the best of our knowledge, we are the first work to propose a task-aware token pruning method for unified transformers, effectively reducing computation and memory usage while maintaining or improving performance.

## 2 RELATED WORK

### 2.1 UNIFIED MULTI-MODAL TRANSFORMERS

Recently, there are several research work (Ge et al., 2024b; Wu et al., 2023; Team, 2024; Xie et al., 2024; Liu et al., 2024b; Sun et al., 2023; Zhou et al., 2024; Tang et al., 2024; Dong et al., 2024; Wang et al., 2024; Ma et al., 2024; Liu et al., 2024a; Wu et al., 2024a; Shi et al., 2024; Anil et al., 2023; Qu et al., 2024; Li et al., 2024a; Kou et al., 2024; Li et al., 2024b) focusing on unified transformers that are capable of both generation and understanding. Chameleon (Team, 2024) and Emu3 (Wang et al., 2024) employs an autoregressive approach for both generation and understanding tasks. SEED-X (Ge et al., 2024b) introduces a unified system for multimodal understanding and generation, incorporating a diffusion model alongside an LLM for generation tasks. Transfusion (Zhou et al., 2024) utilizes discrete tokens to represent texts and continuous embeddings to represent images. It integrates continuous diffusion methods with autoregressive approaches to handle generation and understanding tasks effectively. Show-o (Xie et al., 2024) adopts discrete diffusion for generation and employs an autoregressive mode for understanding within a single model. JanusFlow (Ma et al., 2024) combines the flow matching method for generation with the autoregressive method for understanding. However, all of these models require substantial resources for training.

### 2.2 SPARSE COMPUTATION FOR TRANSFORMERS

**Language Only.** Recently, Large Language Models (LLMs) have developed rapidly, resulting in the need for ever-increasing computational resources (Touvron et al., 2023; Li et al., 2023b; Abdin et al., 2024; Brown et al., 2020; Radford et al., 2018). As a result, much research (Ge et al., 2024a; Elhoushi et al., 2024a; Chen et al., 2024c; Xiao et al., 2024) focuses on sparse computation in LLMs. Mixture of Experts (MoE)(Cai et al., 2024; Xue et al., 2024; Dai et al., 2024; Jiang et al., 2024) is a popular method that replaces the feed-forward (FFN) layers of transformer blocks with MoE layers, where input tokens are dynamically processed by the top-K experts via a router. However, MoE does not reduce training costs and is only efficient during inference. Mixture of Depths (MoD)(Raposo et al., 2024) adopts a router at interleaved layer to decide whether tokens bypass the entire layer, thereby enhancing computational efficiency. Besides, there are several work using skip layers or early exit for sparse computation for LLMs (Elhoushi et al., 2024b; Del Corro et al., 2023; Geva et al., 2022)

**Multimodal understanding and generation.** MoE-LLaVA (Lin et al., 2024a) and $\gamma$-MoD (Luo et al., 2024) investigate Mixture of Experts (MoE) and Mixture of Depths (MoD) in multimodal large language models (MLLMs)(Liu et al., 2024e;c; Bai et al., 2023; Liu et al., 2024d), with $\gamma$-MoD introducing the ARank metric to assess token redundancy per layer. VideoLLM-MoD(Wu et al., 2024b) applies MoD to video LLMs (Chen et al., 2024a). MoMa (Lin et al., 2024b) integrates MoE and MoD into the Chameleon model (Team, 2024). However, it lacks results on generation tasks and most understanding benchmarks. Additionally, its application of MoD involves only a simplistic combination, without a design tailored for unified transformers. In the generation domain, concurrently, several studies (You et al., 2024; Shen et al., 2024; Sun et al., 2024; Fei et al., 2024) have explored MoD or MoE methods for continous diffusion transformer models (Peebles & Xie, 2023). In our work, we apply MoD to various unified transformers, demonstrating comparable or even improving results with fewer computational resources.

## 3 EMPIRICAL ANALYSIS OF UNIFIED TRANSFORMERS

In this section, we first introduce the unified transformers (Xie et al., 2024; Wang et al., 2024) and the Mixture of Depths (MoD)(Raposo et al., 2024) (Sec.3.1). We then separately analyze the attention weights (Sec.3.2), evaluate token redundancy in unified transformers (Sec.3.3), and analyze the interactions across different tasks (Sec.3.4). While Show-o serves as the main model for our experiments, we also investigate variations in other unified transformers (Ma et al., 2024; Liu et al., 2024a; Wang et al., 2024) to gain a more comprehensive insight.

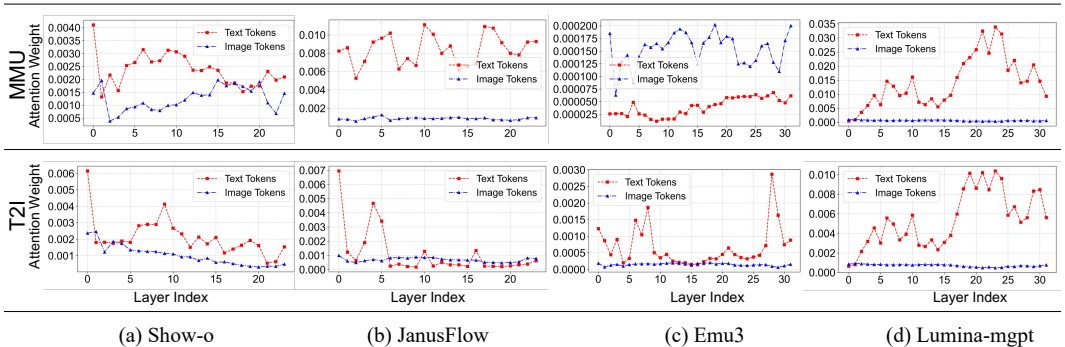

(a) Show-o        (b) JanusFlow        (c) Emu3        (d) Lumina-mgpt

Figure 2: Attention weight for text and image tokens across different transformer layers for two tasks: Multi-Modal Understanding (MMU) and Text-to-Image generation (T2I). Each curve represents one token type, showing how attention allocation changes with model, layer index, and task.

## 3.1 PRELIMINARY

**Unified multimodal transformers**. Unified transformers typically come in two main types: one where both the generation and understanding tasks use AR-based methods, and the other where the generation task uses diffusion or flow matching while the understanding task employs autoregressive methods. We consider two representative unified multimodal transformer models: **Show-o** (Xie et al., 2024) and **Emu3** (Wang et al., 2024). In Show-o, we write the image token set as $\mathbf{u} = \{u_1, \ldots, u_M\}$ and the text token set as $\mathbf{v} = \{v_1, \ldots, v_N\}$. It uses next token prediction (NTP) for image understanding and mask token prediction (MTP) for image generation:

$$\mathcal{L}_{\text{NTP}} = \sum_{i=1}^{N} \log p_\theta(v_i \mid v_{1:i-1}, u_{1:M}), \quad \mathcal{L}_{\text{MTP}} = \sum_{j=1}^{M} \log p_\theta(u_j^* \mid u_{1:j-1}, u_{j+1:M}, v_{1:N}). \quad (1)$$

Emu3 uses only the NTP objective, predicting every next token (image or text).

**Mixture of depths (MoD)** (Raposo et al., 2024). MoD is an efficient training method that reduces the number of tokens processed in each layer of a transformer. Each layer employs a router that decides which tokens will be processed by that block. Tokens that are not chosen will skip the layer and proceed to the next layer. Thus, MoDs can be written as

$$x_i^{l*} = \begin{cases} x_i^l + D^l(x_i^l) R^l(x_i^l), & \text{if } R^l(x_i^l) \geq \delta_s^l, \\ x_i^l, & \text{if } R^l(x_i^l) < \delta_s^l. \end{cases} \quad (2)$$

Formally, let $x$ denote the input sequence of tokens, where each token is represented as a vector $x_i \in \mathbb{R}^d$ and let $\delta_s^l$ be the routing threshold at each layer. $D^l$ represents the $l$-th layer of the Transformer. $R^l$ is the corresponding routing function. Tokens that are not chosen will skip the layer, reducing computational cost. While MoE (Dai et al., 2024) increases model capacity by activating a subset of experts per token, it does not directly lower training cost. In contrast, MoD reduces computation by skipping unnecessary tokens at each layer. Thus, we adopt MoD to train unified transformers more efficiently.

## 3.2 ATTENTION WEIGHTS

We analyze the differences in attention weight patterns across tasks and modalities in unified transformers. We choose four unified models—Show-o (Xie et al., 2024), JanusFlow (Ma et al., 2024), Emu3 (Wang et al., 2024) and Lumina-mgpt (Liu et al., 2024a)—to calculate the average attention weights received by image and text tokens across various layers. The formula and computation steps are detailed in Sec.A.7. Based on the experimental results in Fig. 2, we draw a key observation regarding the attention weight distribution.

**Observation 1**: *Attention weight patterns of different modalities show significant differences depending on the task.*

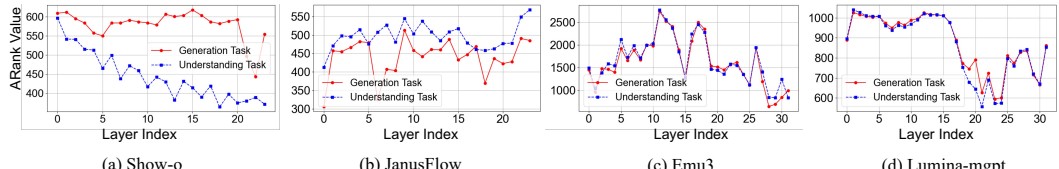

(a) Show-o     (b) JanusFlow     (c) Emu3     (d) Lumina-mgpt

Figure 3: ARank variations across different layers for four unified transformers: Show-o, JanusFlow, Emu3, and Lumina-mgpt. ARank, defined as the rank of the attention map, represents sequence redundancy within each layer. Higher ARank values indicate lower sequence redundancy.

As shown in Fig. 2, for Show-o, JanusFlow and Emu3, the attention weight patterns differ significantly between tasks. However, in Lumina-mgpt, the attention weight patterns are similar across both tasks.

Different tasks in unified transformers often adopt distinct modeling and sequence designs. Show-o and JanusFlow use diffusion or flow-matching for generation and autoregression for understanding, while Emu3 applies autoregression to both but varies in sequence organization, affecting modality performance. In contrast, Lumina-mgpt employs interleaved training with consistent design, yielding similar attention patterns across tasks.

Through observing the attention weight patterns, we realize that the importance of image and text tokens varies across tasks. Therefore, during pruning, we consider redundancy in tokens from all modalities, making the goal to prune tokens across both image and text modalities.

### 3.3 LAYER IMPORTANCE AND TOKEN REDUNDANCY

We explore the significant variation in the importance of different layers and token redundancy for each task from two perspectives. First, we conduct simple inference experiments to analyze the benchmark performance in the Show-o model. Second, we use the ARank metric (Luo et al., 2024) to evaluate token redundancy across layers in different unified transformers.

In the Show-o model, during the inference stage, we skip the odd-numbered layers and evaluate its performance on the GQA benchmark (Hudson & Manning, 2019), as shown in Tab.1 and we draw one observation.

**Observation 2**: *The contribution of each layer to the final outcome is different.*

As shown in Tab. 1, GQA performance declines more significantly when tokens are skipped in the early layers compared to the late layers. This indicates that early layers are more critical for achieving optimal results.

Table 1: GQA performance metrics observed when each specific layer is skipped during inference.

| layer | 1 | 3 | 5 | 7 | 9 | 11 | 13 | 15 | 17 | 19 | 21 | 23 |
|-------|-----|-----|------|------|------|------|------|------|------|------|------|------|
| GQA | 35.0 | 0.0 | 48.0 | 49.0 | 49.4 | 50.0 | 50.2 | 51.7 | 51.5 | 51.4 | 51.1 | 50.9 |

Furthermore, we quantitatively assess the redundancy of tokens within each layer. Drawing inspiration from $\gamma$-mod (Luo et al., 2024), which utilizes the Attention Map Rank (ARank) metric to evaluate token-level redundancy in a layer, we apply this metric to analyze the unified transformers. ARank is calculated as the mean rank of the attention matrices:

$$\tau(x^l, D_l) = \frac{1}{n_h} \sum_{h=1}^{n_h} \text{rank}(A_h), A_h = (x^l W_{Qh})(x^l W_{Kh})^\top. \tag{3}$$

In these equations, $\tau(x^l, D_l)$ denotes the ARank value at the $l$-th layer, where $x^l \in \mathbb{R}^{N \times d}$ is the input with sequence length $N$ and hidden dimension $d$. The operator $\text{rank}(\cdot)$ computes the matrix rank. Each attention head $h \in \{1, \ldots, n_h\}$ has an attention map $A_h^l \in \mathbb{R}^{N \times N}$, obtained from query and key projections with weight matrices $W_{Qh}^l, W_{Kh}^l \in \mathbb{R}^{d \times d_h}$. This metric allows us to quantify the redundancy of tokens in each layer, providing insights into how different tasks and layers contribute to overall model efficiency. A layer with a low ARank means that most of its tokens are less informative. Based on the experimental results as shown in Fig. 3, we draw two observations regarding the token redundancy between different tasks.

**Observation 3**: *The number of redundant tokens differs significantly between tasks with divergent modeling methods.*

As illustrated in Fig. 3, ARank values differ between tasks in the Show-o and JanusFlow models. In Show-o, the Text-to-Image(T2I) generation sequence has significantly higher ARank values than the Multi-Modal Understanding(MMU) sequence, indicating more redundant tokens in the MMU task. Conversely, Lumina-mgpt and Emu3 exhibit similar redundancy levels across both tasks. We attribute this difference to their modeling approaches: Show-o and JanusFlow use diffusion or flow matching for generation and autoregressive models for understanding, while Lumina-mgpt and Emu3 employ autoregressive methods for both tasks.

**Observation 4**: *The redundancy of token sequences differs significantly across layers for different tasks.*

As shown in Figs.3, ARank values are significantly higher in the early layers compared to the later layers. In the Show-o model, the ARank value for the MMU task decreases as the layer index increases. In the T2I task, ARank values also show a substantial decrease in the later layers. All three models—JanusFlow, Lumina-mgpt, and Emu3—exhibit variations in ARank values across layers, indicating differing levels of token redundancy within each layer. We speculate these variations across layers may be related to the language models used.

Based on the analysis of layer importance and token redundancy, we conclude that pruning should focus on tokens in layers with high redundancy. The ARank metric can be used to identify such layers and determine the proportion of tokens to prune for both tasks.

## 3.4 INTERACTIONS BETWEEN TASKS

**Task interactions in unified transformers.** Using the Show-o model, we test whether joint training affects performance by comparing single-task and multi-task settings. Evaluation on POPE (Li et al., 2023a), MME (Fu et al., 2023), GQA (Hudson & Manning, 2019), and GenEval (Ghosh et al., 2023) shows comparable results across all configurations (Tab. 2). From these observations, we draw a key conclusion: **A large language model can effectively accommodate both tasks, as simultaneous training yields results comparable to training each task individually in the Show-o model.** This lack of mutual enhancement likely stems from the Show-o model using discrete diffusion for generation and autoregressive processing for understanding.

Table 2: Benchmark results for three training configurations: Show-o* (training T2I and MMU tasks), Only T2I, and Only MMU.

| Method | MME↑ | GQA↑ | POPE↑ | GenEval↑ |
|---|---|---|---|---|
| Show-o* | 1032.0 | 52.5 | 77.9 | 0.63 |
| Only MMU | 1030.0 | 52.2 | 77.7 | – |
| Only T2I | – | – | – | 0.63 |

**Competitive token pruning between two tasks.** We design a competitive framework where Text-to-Image (T2I) and Multi-Modal Understanding (MMU) tokens vie for selection, enabling comparison of their importance. Using Straight-Through Gumbel-Softmax (Sec. A.6), tokens receive binary weights for pruning. To limit redundancy, we set router capacity to 0.5 and add an auxiliary loss, so each layer processes only half of the tokens. The formula is provided in the Sec. A.6 and Fig. 4 illustrates the outcome.

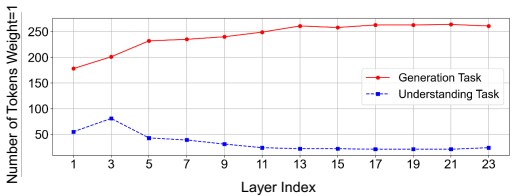

Figure 4: Token weight assignment using Gumbel Softmax. A higher number of tokens assigned a weight of 1 across layers indicates greater importance of generation task tokens compared to understanding task tokens.

**Observation 5**: *In the Show-o model, generation tokens are assigned higher weights, indicating that they contribute more significantly to the final results compared to understanding tokens.*

Fig. 4 shows that T2I tokens are almost always retained, while MMU tokens are pruned more aggressively, indicating T2I tokens contribute more to loss reduction. Treating all tokens uniformly causes imbalance, as one task dominates. Results from Show-o suggest minimal cross-task enhancement, implying that task-specific pruning may yield more balanced and effective performance.

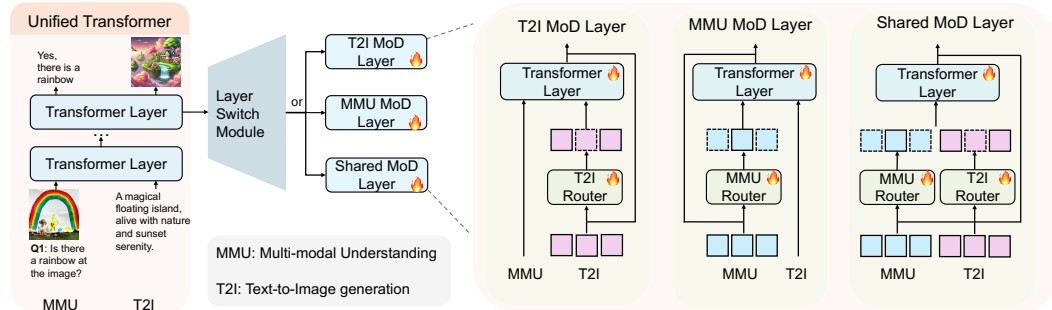

Figure 5: Pipeline of UniMoD. The Layer Switch Module transforms dense transformer layers into three specialized types: T2I MoD layers for Text-to-Image (T2I) generation, MMU MoD layers for Multi-Modal Understanding (MMU), and Shared MoD layers for both tasks. For each task, task-aware routers with distinct capacities prune tokens of different modalities, thereby enhancing computational efficiency and maintaining performance across tasks.

## 4 METHOD

### 4.1 TASK-AWARE MIXTURE-OF-DEPTHS

Based on our experimental results and analyses, we observe that token redundancy varies across tasks and layers. By analyzing the attention weights of unified transformers (see Sec. 3.2), we find that the importance of image and text tokens varies across tasks. It suggests that token pruning should target tokens from all modalities. Our experiments with ARank values (Sec.3.3) across various layers and tasks show that redundancy levels depend on both the task and the specific layer. It suggests using the ARank metric to identify layers for pruning. Additionally, our task competition experiments (Sec. 3.4) reveal that token importance varies across tasks in terms of final loss optimization. It suggests tokens should be pruned separately for each task. Consequently, we introduce a task-aware token pruning method. The whole pipeline of UniMoD is shown in Fig. 5

**Task-aware MoD layer.** We transform dense transformer blocks into three specialized MoD blocks: the T2I MoD block for pruning tokens in the Text-to-Image (T2I) generation task, the MMU MoD block for pruning tokens in the Multi-Modal Understanding (MMU) task, and the Shared MoD block for pruning tokens in both tasks, as shown in Figure 5. Specifically, the T2I MoD block only prunes T2I tokens while processing all MMU tokens, and the MMU MoD block only prunes MMU tokens while processing all T2I tokens. The Shared MoD block simultaneously prunes tokens from both tasks. For each task, we design dedicated routers with specific capacities, enabling the model to adaptively prune tokens based on task requirements. These routers target tokens from different modalities, ensuring efficient multi-modal processing and enhancing the model's overall computational efficiency.

**Layer switch module.** We use the ARank metric to determine which transformer layers should be converted into MoD blocks. The procedure consists of three steps. (1) *Layer selection.* For each layer in the Show-o model, we compute ARank across different tasks using 50 samples per task. From the ARank line chart, we select the half of layers with the lowest values for each task. (2) *Pruning ratio estimation.* We approximate each layer's pruning ratio by normalizing its ARank score by the sequence length, which enables task-specific pruning that reflects token redundancy. (3) *Token pruning.* Following the MoD procedure, each router assigns scores to tokens and retains the Top-K tokens with the highest scores, where $K$ is determined by the pruning ratio computed in step (2).

To accommodate different tasks, we design task-specific routers for each task (e.g., T2I and MMU) to perform different token pruning strategies. We also set task-specific thresholds. The formula is

$$x_i^{*l} = \begin{cases} x_i^l + D_t^l\left(x_i^l\right) \ R_t^l\left(x_i^l\right), & \text{if } R_t^l\left(x_i^l\right) \geq \delta_t^l, \\ x_i^l, & \text{if } R_t^l\left(x_i^l\right) < \delta_t^l. \end{cases} \tag{4}$$

$x_i$ is the original token before pruning and $x_i^*$ is the updated token after applying the pruning strategy. $l$ is the layer index of the transformer and $t$ represents the specific task (e.g., T2I, MMU). $D_t(x_i)$ is

the task-specific router function for task $t$, determining how the token $x_i$ should be adjusted. $D_t(\cdot)$ is the corresponding routing function and $R_t(x_i)$ is the task-specific weight for token $x_i$ in task $t$, indicating the token's relevance or significance. $\delta_t$ is the task-specific threshold for task $t$, deciding whether the token $x_i$ should undergo pruning based on its importance score. This allows us to apply different token pruning criteria for each task, enabling dynamic adjustment of token pruning based on the specific requirements and ARank values of each task.

## 5 EXPERIMENTS

### 5.1 IMPLEMENTATION DETAILS

We choose Show-o (Xie et al., 2024) and Emu-3 (Wang et al., 2024) as representative models for our approach to token pruning, as they cover different types of unified transformers. Show-o integrates the diffusion pipeline with an autoregressive mode, addressing both generative and understanding tasks using distinct mechanisms. In contrast, Emu-3 employs the autoregressive mode for both tasks. By selecting these two models, we demonstrate that our method is applicable to a wide range of unified models.

In the Show-o model, when selecting layers to convert into MoD (Mixture of Depths) layers, we transform the last 12 layers into MoD layers for both tasks. For the Multi-Modal Understanding (MMU) task, we scale the capacity from 1 down to 0.2. For the Text-to-Image (T2I) task, we prune 20% of the tokens in the later layers. We use a batch size of 10 for both the MMU and T2I tasks. For the T2I task, we use the image datasets from the original Show-o model, and for MMU, we employ the Cambrian dataset (Tong et al., 2024). Since Emu3 does not release MMU training resources, we use the LLaVA-v1.5-mix-665K dataset and add MMU-specific code, while keeping the same T2I data as Show-o. The model is finetuned on 8 H100 GPUs with 80% token pruning in the last 16 layers. Further dataset and implementation details are in Sec. A.1.

Table 3: Comparison of UniMoD with baseline methods across Multi-Modal Understanding (MMU) and Text-to-Image (T2I) benchmarks in the Show-o and Emu3 Models.

| | TFLOPS↓ | MME↑ | GQA↑ | POPE↑ | MMMU↑ | VQAv2↑ | GenEval↑ | DSG↑ | CLIP_score↑ |
|---|---|---|---|---|---|---|---|---|---|
| Show-o | 51.1 | 1056.0 | 56.3 | 79.8 | 25.8 | 68.3 | 0.62 | 72.2 | 0.331 |
| Interleaved Layer | 25.6 | 828.4 | 45.6 | 74.2 | 27.2 | 53.9 | 0.29 | 15.6 | 0.302 |
| EarlyExit | 25.6 | 947.0 | 51.1 | 79.5 | 24.1 | 60.4 | 0.26 | 56.2 | 0.294 |
| UniMoD | 43.3 | 1093.7 | 54.5 | 80.3 | 25.7 | 66.2 | 0.61 | 73.6 | 0.332 |
| Emu3 | 89.0 | 881.3 | 46.0 | 76.0 | 25.1 | 54.8 | 0.46 | 79.0 | 0.318 |
| UniMoD | 53.5 | 901.0 | 45.2 | 74.7 | 25.3 | 53.9 | 0.48 | 80.0 | 0.321 |

### 5.2 MAIN RESULTS

To evaluate multimodal understanding, we use the POPE(Li et al., 2023a), MME (Fu et al., 2023), VQAv2 (Goyal et al., 2017), GQA (Hudson & Manning, 2019), and MMMU (Yue et al., 2024) benchmarks. To evaluate generation capabilities, we use the GenEval (Ghosh et al., 2023), DSG-1K (Cho et al., 2024), and CLIP-

Table 4: Training Cost Comparison.

| Model | Params | T2I | | MMU | |
|---|---|---|---|---|---|
| | | TFLOPs | Training Cost | TFLOPs | Training Cost |
| Show-o | 1.4B | 51.1 | 1.30s/iter & 67G | 51.1 | 1.30s/iter & 67G |
| UniMod | 1.4B | 45.9 | 1.27s/iter & 64G | 40.8 | 1.25s/iter & 61G |
| Emu3 | 8.5B | 89.0 | 3.56s/iter & 65G | 89.0 | 3.56s/iter & 65G |
| UniMod | 8.5B | 53.5 | 2.80s/iter & 64G | 53.5 | 2.80s/iter & 64G |

Score (Radford et al., 2021) benchmarks. To calculate the CLIP score, we use 2,000 generated images sampled from the GenEval and DSG-1K benchmarks. To assess efficiency, we measure TFLOPs and training speed. Following the practice in DiT (Peebles & Xie, 2023), we estimate the training compute as model TFLOPs × batch size × 3. The factor of 3 approximates the backward pass as requiring twice the compute of the forward pass.

**Baselines.** We compare our approach with various baselines, as shown in Tab.3. *Full Computation.* Using the vanilla training pipeline, all visual tokens and text tokens are passed through all transformer layers without any pruning. *Early Exit.* Building on studies of language-only LLMs (Geva et al., 2022), we adopt a similar method for our unified models by performing an early exit at the 12th layer. *Interleaved Layer Skipping.* We apply a skipping-layers method, removing every token in the interleaved layers. This is equivalent to our method with a capacity of 1 in odd-numbered layers and a capacity of 0 in even-numbered layers.

As shown in Tab. 3, although two baselines consume fewer TFLOPs, their performance is significantly inferior to the full computation model, especially for the T2I task. In contrast, our method achieves the best balance between performance and efficiency. We reduce the training 15% FLOPs while maintaining comparable performance and even attaining better results on some benchmarks in the Show-o model. As shown in Tab. 3, in the Emu model, our method reduces FLOPs by 40%, while achieving comparable or better results compared to the full computation model in both T2I and MMU tasks. Our full Emu3 results differ from the original paper because we use alternative training datasets, as the official code and data are not publicly available. A comparison with MoE is provided in Sec. A.9, and additional Emu3 results are reported in Sec. A.8.

**Improved efficiency with larger model scale.** Our method grows more efficient as model size increases. Training Show-o on the Llama 8B base model yields a 20% reduction in FLOPs compared with 15% in the 1.3B model. The detailed results for the 8B model are presented in Sec. A.3.

**Training cost.** As shown in Tab. 4, our method not only reduces FLOPs but also improves memory usage and training speed. In the Emu3 model, our approach is more efficient than Show-o in terms of FLOPs and training speed. We attribute this difference to the design of their image tokenizers: Emu3 uses 4096 tokens per image, while Show-o uses 1024 tokens. More tokens in Emu3 introduces more redundancy, enhancing the efficiency of our method. The improvement in memory usage is less significant due to Emu3's larger model size. Specific reasons are further discussed in the Sec. A.2.

**Pareto frontier analysis during inference.** We analyze the trade-off between token pruning and inference performance in Show-o. The pruning ratio chosen during training effectively reduces tokens with minimal performance loss during inference. Detailed results are in Sec. A.4.

**Visualization results.** We present visualization examples for both tasks in Sec. A.10. UniMoD achieves comparable or better results in both quantitative and visual evaluations.

**Scaling to more than two tasks.** Our method naturally extends beyond two tasks. As an example, we analyze pure text tasks, with details and results provided in Sec. A.12, confirming the scalability of our approach.

**Adaptation to diffusion models.** While our method is primarily designed for unified transformers, we also demonstrate its effectiveness in training and fine-tuning generation models such as DiT (Peebles & Xie, 2023) and PixArt (Chen et al., 2024b). Experimental results and implementation details are provided in the Sec. A.5.

## 5.3 ABLATION STUDIES

To assess the effect of each design element in UniMoD, we carry out an ablation study on the Show-o model, as shown in Tab. 5. We compare three variants: (1) Basic MoD, where MoD is directly added to the unified trans-

Table 5: Ablation studies of UniMoD in the Show-o model.

| Method | TFLOPS↓ | MME↑ | GQA↑ | POPE↑ | MMMU↑ | VQAv2↑ | GenEval↑ |
|---|---|---|---|---|---|---|---|
| Basic MoD | 40.8 | 960.6 | 51.2 | 76.9 | 23.9 | 63.2 | 0.15 |
| w/o layer switch module | 43.3 | 920.3 | 52.1 | 74.7 | 25.1 | 63.2 | 0.50 |
| w/o task-aware router | 40.8 | 1052.0 | 54.4 | 80.2 | 25.6 | 65.5 | 0.50 |
| UniMoD | 43.3 | 1093.7 | 54.5 | 80.3 | 25.7 | 66.2 | 0.61 |

former; (2) without the layer switch module, which prunes tokens only in the interleaved layers; and (3) without the task-aware router, which uses a single router to prune tokens for both tasks while still selecting specific layers. The Basic MoD variant produces the lowest performance on both tasks. Using separate routers at interleaved layers slightly improves generation compared to basic MoDs, but outcomes remain suboptimal. Employing a single router for both tasks at specific layers slightly worsens understanding results and severely degrades generation performance. In contrast, our method achieves superior performance by effectively balancing efficiency and effectiveness. To ensure fairness, each ablation experiment maintains the same pruning rate as our method.

## 6 CONCLUSION

In this work, we present an efficient training method for unified transformers by analyzing attention weights, layer importance, and task interactions to identify sequence redundancy. Using these insights, we introduce a task-aware token pruning approach that reduces FLOPs while maintaining or enhancing performance across various benchmarks. Our method effectively balances efficiency and performance, demonstrating its applicability to unified transformers.

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

## A   TECHNICAL APPENDICES

In this appendix, we provide additional implementation details (Sec. A.1), discussion of training cost (Sec. A.2), results of adapting MoD to diffusion models (Sec. A.5), the Straight-Through Gumbel Softmax formula (Sec. A.6), and the attention weight formula (Sec. A.7). We further report Show-o results with an 8B LLM (Sec. A.3), Emu3 experiments (Sec. A.8), MoE-related baselines (Sec. A.9), and scaling to more than two tasks (Sec. A.12). We also include visualization results (Sec. A.10), a description of our use of LLMs (Sec. A.11), and a discussion of limitations (Sec. A.13).

### A.1   MORE IMPLEMENTATION DETAILS

**Dataset enhancement and streamlined training workflow in the Show-o Model.** The original Show-o model consists of multiple training stages. In the first two stages, it is trained on the ImageNet (Deng et al., 2009) dataset and large-scale text-image paired data to achieve effective text-image alignment. The third stage leverages high-quality data to develop generation capabilities, while the final two stages utilize the LLaVA dataset (Liu et al., 2024e) to enhance understanding capabilities.

In this work, we improve the Show-o training pipeline by incorporating additional understanding datasets and reducing the training process to two stages. The original Show-o exclusively used the LLaVA dataset (Liu et al., 2024e) for training its understanding component. However, data imbalance caused the model to develop generation capabilities before understanding capabilities, which adversely affected generation quality. To resolve this, we introduce the Cambrian dataset (Tong et al., 2024) and internal high-quality data to fine-tune the Show-o model within a two-stage training framework. The model processes images at a resolution of 512×512 pixels. Our revised pipeline achieves comparable results while reducing computational resource usage. We reevaluate the primary benchmarks and compare them with the original Show-o results.

**Emu3.** The Emu3 public repository does not include the training code or data for MMU tasks. To address this, we utilize the LLaVA-v1.5-mix-665K dataset to represent MMU capabilities and enhance the training pipeline with additional MMU-specific code. We employ the same generation data as the Show-o model and fine-tune Emu3 for 2 epochs with a learning rate of 2e-5, processing images at a resolution of 512×512 pixels.

In our experiments, we use the Show-o and Emu3 checkpoints from the first training stage, which involves low-quality text-to-image generation and limited captioning capabilities, as base models. To validate our method, we fine-tune these models with high-quality image data and understanding QA data. The router utilizes a single linear network.

### A.2   DISCUSSION ON TRAINING COSTS

As shown in Tab. 4, although Emu3 prunes more tokens and achieves greater reductions in FLOPs and training speed, its memory usage remains largely unchanged compared to the full computation model. This discrepancy arises from the significant size difference between Emu3 and Show-o, with Emu3 having 8.5B parameters compared to Show-o's 1.4B. In Emu3, the large number of parameters and their corresponding gradients dominate memory consumption, making token pruning's impact on memory minimal. Instead, pruning primarily enhances training speed because the extensive parameter count leads to substantial attention computation per layer, making token reduction more impactful on speed. In contrast, the smaller Show-o model allocates a substantial portion of memory to activation variables during token computation. Consequently, token pruning in Show-o leads to a more significant reduction in memory usage.

### A.3   SHOW-O IN LLAMA 8B MODEL

To further validate the potential of our method, we train Show-o on the Llama3 8B model. In this setting, we achieve a 20% reduction in TFlops, compared to a 15% reduction in the Phi 1.3B model, as shown in Tab. 6

We can see that using a larger model improves the performance of both the original Show-o model and our method. Moreover, our method reduces more TFlops, validating its potential.

Table 6: Comparison of TFLOPS and benchmark performance for Show-o(8B) and UniMoD.

| Method | TFLOPS | MME | GQA | POPE | VQAv2 | MMMU | GenEval |
|--------|--------|-----|-----|------|-------|------|---------|
| Show-o(8B) | 29.6 | 1241 | 59.7 | 80.9 | 70.9 | 26.2 | 0.60 |
| UniMoD | 23.5 | 1223 | 58.6 | 81.6 | 69.8 | 25.6 | 0.58 |

## A.4 PARETO FRONTIER ANALYSIS OF SHOW-O

Pareto Frontier Analysis is a method used to evaluate the trade-off between two competing objectives. In our context, it is used to show how inference performance (e.g., accuracy or reward) changes relative to computational cost (e.g., FLOPs). A point is said to be Pareto-optimal if no other point achieves better performance with lower cost.

To illustrate the trade-off between FLOPs and performance during inference stage, we conduct more inference experiments of Show-o and present these results. As shown in Tab. 8 and Fig. 7, for MMU, we observe that a 15%-20% TFlops reduction—roughly matching the training reduction—is the balance point. Keeping all tokens or pruning too few slightly degrades performance, while excessive pruning worsens results. For T2I, a 10% TFlops reduction (approximately equal to the training reduction) is optimal. Note that few MoD works explore the Pareto Frontier by adjusting the ratio during inference. The experimental results in the tables represent our preliminary exploration. We believe this is a promising direction for future MoD research.

Table 7: Trade-off between TFlops ratio and GQA/POPE performance.

| TFlops ratio | 0% | 15% | 20% | 25% | 35% |
|--------------|-----|------|------|------|------|
| GQA | 53.5 | 54.4 | **54.5** | 54.1 | 50.7 |
| POPE | 80.0 | **80.4** | 80.2 | 79.9 | 77.0 |

Table 8: Trade-off between TFlops ratio and GenEval performance.

| TFlops ratio | 0% | 10% | 15% | 20% |
|--------------|-----|------|------|------|
| GenEval | 0.607 | **0.610** | 0.572 | 0.523 |

## A.5 ADAPTATION TO DIFFUSION MODELS

**MoD for Generation Models.** While our method is primarily designed for unified transformers, we also validate its effectiveness for training or fine-tuning generation models such as DiT (Peebles & Xie, 2023) and PixArt (Chen et al., 2024b). For the DiT model, we select the checkpoint trained for 50k iterations as the base model to calculate the ARank. For the PixArt model, we use the final model as the base and fine-tune it using our internal high-quality data (used in the Show-o fine-tuning stage). We evaluate the DiT model using the ImageNet (Deng et al., 2009) FID metric, and for PixArt, we use the GenEval benchmark (Ghosh et al., 2023) to assess generation quality. Both models process images at a resolution of 256×256 pixels.

Table 9: UniMoD for PixArt. UniMoD achieves results closest to the full computation model.

| Model | Method | TFLOPS↓ | Single Obj. | Two Obj. | Counting | Colors | Position | Color Attri. | Overall↑ |
|-------|--------|---------|-------------|----------|----------|--------|----------|--------------|----------|
| | Original Model | 42.64 | 0.98 | 0.50 | 0.44 | 0.80 | 0.08 | 0.07 | 0.48 |
| PixArt-alpha | Full Computation | 42.64 | 0.98 | 0.71 | 0.55 | 0.82 | 0.17 | 0.29 | 0.58 |
| | Interleaved Layer | 32.03 | 0.98 | 0.59 | 0.43 | 0.80 | 0.13 | 0.22 | 0.53 |
| | UniMoD | 32.03 | 0.99 | 0.68 | 0.47 | 0.81 | 0.17 | 0.26 | 0.56 |

For PixArt, we conduct three experiments. First, we present the results using full computation. Second, we prune 40% of the tokens in the interleaved layers. Finally, we calculate the ARank value for each layer and select the 14 layers with the lowest ARank values to prune 40% of the tokens. As shown in Tab. 9, our ARank-based MoD method achieves better performance compared to standard pruning approaches at the same computational cost, with only a slight reduction in performance relative to full computation.

For DiT, we select the checkpoint trained for 50k iterations as the base model to calculate the ARank. Then we choose the least value 14 layers to prune tokens. In practice, we gradually scale the token capacity from 1 to 0.2 over the course of 500k training iterations. Finally, we evaluate the model using a capacity of 0.4, as shown in Tab. 10.

From these experiments, we observe that the token sequences in both DiT and PixArt exhibit higher redundancy compared to those in Show-o. We attribute this difference to the design of their image tokenizers. The VAEs used in DiT and PixArt downsample images by a factor of 8, whereas Show-o's tokenizer downsamples by a factor of 16. Consequently, for images of the same resolution, DiT and PixArt require more tokens to represent the image than Show-o.

Table 10: UniMoD for DiT. UniMoD achieves similar results as the original model after 500K iterations.

| Model | Method | TFLOPS↓ | IS↑ | FID↓ |
|-------|--------|---------|-----|------|
| DiT-XL/2 | Full Computation | 117.2 | 168.16 | 4.97 |
| | UniMoD | 93.6 | 171.67 | 5.45 |

### A.6 STRAIGHT-THROUGH GUMBEL SOFTMAX

The Straight-Through Gumbel-Softmax method assigns binary weights to tokens, allowing discrete sampling while preserving differentiability through a straight-through estimator essential for gradient-based optimization. In the forward pass, tokens with the highest probability are selected for retention or pruning. During the backward pass, soft probabilities are used to compute gradients, allowing smooth updates.

The formula for the Gumbel Softmax is:

$$y_i = \frac{\exp\left(\dfrac{\log(\pi_i) + g_i}{\tau}\right)}{\sum\limits_{j=1}^{K} \exp\left(\dfrac{\log(\pi_j) + g_j}{\tau}\right)}. \tag{5}$$

$\pi_i$ is the original probability of the $i$-th category, $g_i$ is noise sampled from the Gumbel$(0, 1)$ distribution, $\tau$ is the temperature parameter, and $K$ is the number of categories.

In the straight-through version, we obtain a hard one-hot vector z during the forward pass by taking the index with the highest y:

$$z_i = \begin{cases} 1, & \text{if } i = \arg\max_j y_j \\ 0, & \text{otherwise} \end{cases} \tag{6}$$

In this work, we employ this method to assess the importance of tokens across different tasks. The auxiliary loss is defined as $\mathcal{L}_{\text{aux}} = P \sum_i (r_i - P)^2$, where $r_i$ is the capacity of the $i$-th layer ($i \leq L$). This loss function is utilized to ensure that approximately half of the tokens are processed by each layer.

### A.7 ATTENTION WEIGHT FORMULA

In a transformer attention layer, let us denote the attention weight matrix by

$$A = [\alpha_{ij}] \in \mathbb{R}^{n \times n},$$

where represents the attention score from token j receives an average attention from all other tokens, which is calculated as

$$r_j = \frac{1}{n-1} \sum_{\substack{i=1 \\ i \neq j}}^{n} \alpha_{ij}, \quad \text{for } j = 1, 2, \ldots, n,$$

The final attention weight for the image modality is computed by averaging rj over all image tokens, and similarly for the text modality. The formulas are given by

$$\alpha_{\text{final}}^{\text{img}} = \frac{1}{n_I} \sum_{j \in I} r_j = \frac{1}{n_I(n-1)} \sum_{j \in I} \sum_{\substack{i=1 \\ i \neq j}}^{n} \alpha_{ij},$$

$$\alpha_{\text{final}}^{\text{text}} = \frac{1}{n_T} \sum_{j \in T} r_j = \frac{1}{n_T(n-1)} \sum_{j \in T} \sum_{\substack{i=1 \\ i \neq j}}^{n} \alpha_{ij}.$$

For the MMU task, we select 50 samples from the MME, GQA, and POPE benchmarks to compute the attention weights and ARank values shown in Fig. 2 and Fig. 3. For the T2I task, we select 20 prompts from the GenEval benchmark to compute the metrics.

## A.8 MORE EXPERIMENTAL RESULTS OF EMU3 MODEL

**MMU result of finetuning on the final checkpoint.** For the MMU task, in addition, we supplement an experiment on continued fine-tuning based on the final Emu3 checkpoint. We observe improvements in the performance of both Emu3 and our method compared to the results shown in Tab. 3. Moreover, our method achieves comparable performance while reducing TFlops by 40%.

Table 11: MMU task performance of Emu3 and UniMoD under different training settings. Continued fine-tuning improves Emu3, and UniMoD reduces TFLOPS by 40% while matching performance.

| Method | TFLOPS↓ | MME↑ | GQA↑ | POPE↑ | MMMU↑ | VQAv2↑ |
|---|---|---|---|---|---|---|
| Emu3 (old) | 89.0 | 881 | 46.0 | 76.0 | 54.8 | 25.1 |
| Emu3 | 89.0 | 1040 | 52.9 | 78.7 | 63.0 | 25.1 |
| UniMoD | 52.5 | 992 | 50.4 | 79.2 | 61.2 | 25.4 |

**Comparison with baselines.** We also compare Emu3 with two baselines—Interleaved Layer Skipping and EarlyExit. Our method achieves the best trade-off between efficiency and performance.

Table 12: Comparison of TFLOPS and benchmark performance for different methods.

| Method | TFLOPS | MME | GQA | POPE | VQA2 | MMMU | GenEval |
|---|---|---|---|---|---|---|---|
| Emu3 | 89.0 | 881.3 | 46.0 | 76.0 | 54.8 | 25.1 | 0.46 |
| EarlyExit | 53.6 | 830.0 | 44.8 | 73.2 | 53.4 | 26.2 | 0.45 |
| Inteleaved Layer | 52.5 | 718.6 | 38.5 | 65.8 | 47.3 | 25.6 | 0.19 |
| UniMoD | 53.5 | 901.0 | 45.2 | 74.7 | 53.9 | 25.3 | 0.48 |

## A.9 COMPARISON WITH MOE

Mixture-of-Experts (MoE) models and Mixture of Depths (MoD) pursue different objectives. MoE increases model capacity by routing tokens to experts, but this does not reduce token processing and instead raises computational cost. In contrast, MoD skips layers on a per-token basis, directly lowering both training and inference cost. Thus, a direct comparison is not appropriate: MoE increases TFLOPs to boost capacity, whereas MoD reduces TFLOPs for efficiency.

For completeness, we also ran a lightweight MoE experiment on Show-o at $256 \times 256$ resolution with a simple routing setup. Results are shown in Tab. 13.

Table 13: Comparison of Show-o with MoE and UniMoD at $256 \times 256$ resolution. MoE improves capacity but at the cost of significantly higher TFLOPs, while UniMoD achieves better efficiency.

| Method | MME | GQA | POPE | GenEval | TFLOPs |
|---|---|---|---|---|---|
| Show-o (256) | 1032 | 53.5 | 78.0 | 0.62 | 31.0T |
| MoE-based | 1052 | 53.81 | 79.1 | 0.61 | 48.6T |
| UniMoD | 1064 | 52.8 | 78.1 | 0.62 | 23.3T |

## A.10 VISUALIZATION RESULTS

We visualize the results of two tasks for the Show-o model and our UniMoD method. The visualization shows that UniMoD achieves comparable performance to the original model as shown in Fig. A.10.

## A.11 USE OF LLMS

In this work, we employ large language models (LLMs) in two ways. First, LLMs are used to assist with paper writing, such as language polishing and improving clarity of presentation. Second, our

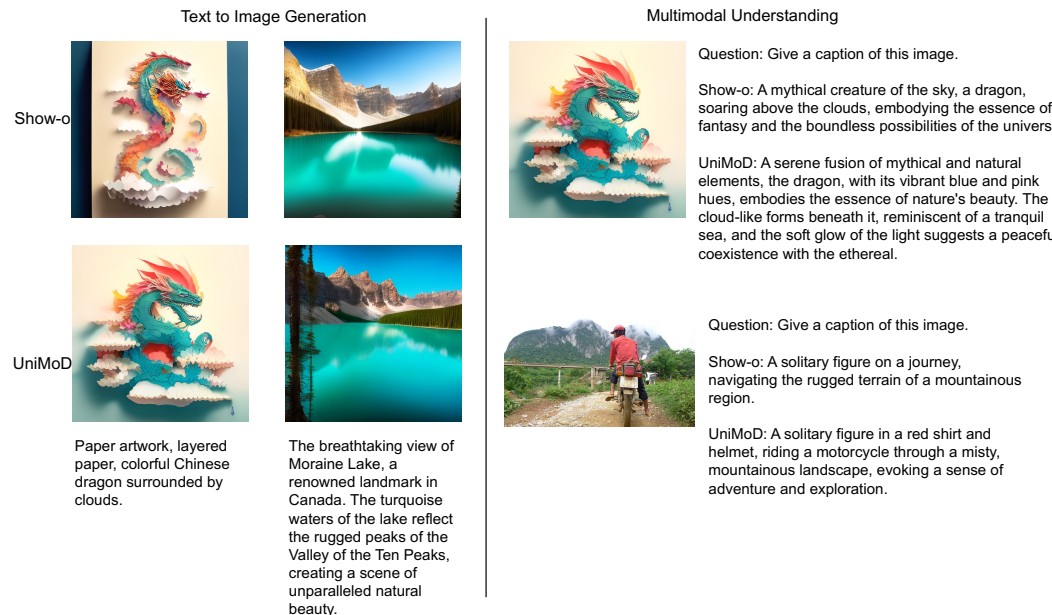

Figure 6: Visualization results of our method and the original model on both generation and understanding tasks.

experiments are conducted on a unified model that is finetuned from an LLM backbone, making LLMs a central component of the model development process.

## A.12  SCALING TO MORE THAN TWO TASKS

Our method can be extended to additional tasks with minimal modification. For each new task, we introduce a separate router, compute ARank on sample sequences, and set the router's keep ratio accordingly.

As an example, beyond image generation (T2I) and multimodal understanding (MMU), we also measure ARank on pure text sequences for the Show-o model. Fig. 7 reports ARank values for all 24 layers across the three tasks. We observe that text ARank patterns diverge significantly from image tasks: beyond the first few layers, text sequences exhibit higher redundancy.

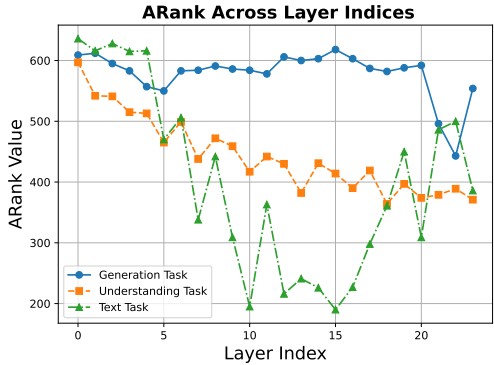

Figure 7: ARank values across 24 layers for generation, understanding, and text tasks. Text shows higher redundancy beyond the early layers.

To further validate scalability to pure-text tasks, we evaluate both the original Show-o and UniMoD on a Wikipedia perplexity benchmark. The baseline Show-o achieves a perplexity of 55.6, while UniMoD reaches 58.9 after only 6k iterations. The absolute gap is 3.3 (about 5.6% relative), which is typically regarded as evaluation noise in language model benchmarks, so we consider the two results

equivalent. Moreover, after only 6k iterations, UniMoD reduces the training loss to 3.0, matching the original approach. These findings confirm that our pruning strategy generalizes effectively beyond image tasks to pure-text scenarios.

## A.13 LIMITATIONS

Our method currently requires calculating the ARank to determine the percentage of pruned tokens. In the future, we hope that MoD can learn the pruning ratio automatically without requiring pre-computation.

