# OpenReview forum: "UniMoD: Efficient Unified Multimodal Transformers with Mixture-of-Depths"
_ICLR.cc/2026/Conference — Submitted to ICLR 2026_

### Official Review · Reviewer_ruma · 2025-10-26

**Soundness:** 2
**Presentation:** 3
**Contribution:** 2
**Rating:** 4
**Confidence:** 4

**Summary:**

This paper proposes UniMoD (Efficient Unified Multimodal Transformers with Mixture-of-Depths) to address the training efficiency issue in existing unified multimodal transformers. A unified multimodal transformer is a model that integrates and processes both generation and understanding tasks within a single model. The observed training inefficiency stems from token redundancy. The paper empirically demonstrated that token redundancy varies across task type, modality, and layer, and proposed UniMoD as a solution. UniMoD addresses the limitations of applying MoD uniformly to unified multimodal transformers by selectively pruning tokens based on task and layer.

**Strengths:**

S1. The paper analyzes the existing token redundancy problem and reveals that it varies across tasks and layers. Based on this finding, it proposes a task-aware token pruning method. The causality between the problem and the proposed modules is clear, and its effectiveness is verified.

S2. The proposed method (UniMoD) experimentally shows that it reduces the computational cost of existing methods while maintaining comparable or even superior performance. For example, it reduces training FLOPs by 15% in Show-o and 40% in Emu3.

S3. The paper considers different modeling approaches of transformers (autoregressive/diffusion) for comparison and verifies the generalizability of UniMoD across various settings.

**Weaknesses:**

W1. Only 2 main models (Show-o and Emu3) are thoroughly evaluated: 1) Show-o: 1.4B params, diffusion + autoregressive,
2) Emu3: 8.5B params, fully autoregressive. Related work mentions MoMa (Lin et al., 2024b) which also integrates MoE/MoD into Chameleon (a unified model). A more direct comparison or clearer differentiation would be beneficial, although the paper notes MoMa lacked results on generation/most understanding tasks.

W2. The method relies on calculating ARank values beforehand to select layers and estimate pruning ratios. This requires running inference on a base model with sample data, adding a preprocessing step and computational overhead before the efficient training can begin.

W3. The pruning ratios appear to be determined statically based on the initial ARank analysis, rather than being learned dynamically during training (though this is acknowledged as a limitation). Performance might be sensitive to the accuracy of this initial estimation.

W4. Although the paper shows a reduction in FLOPs due to the introduction of the proposed UniMoD, it does not provide an analysis of the exact increase in the number of parameters or the computational overhead caused by introducing the method.

**Questions:**

Q1. How much computational overhead does the initial ARank calculation add, and how sensitive are the final results to the number and type of samples used for this calculation?

Q2. Could the task-specific routers or pruning ratios be learned dynamically during training, perhaps conditioned on the task type or layer depth, rather than being fixed based on pre-computed ARank values?

Q3. The proposed method was validated by applying pruning to layers with low ARank values. However, it is necessary to show the contrasting effect when pruning is applied to layers with high ARank values. This experiment would clearly demonstrate the causality between ARank, token redundancy, and the proposed method's effectiveness.

Q4. A clear explanation is needed regarding the experiment parameters (pruning layer location and ratio). For example, in the Show-o model's MMU and T2I scaling and pruning ratio selection, were these decisions entirely based on ARank, or were other heuristic criteria also involved?

---

> ### Author Response · Authors · 2025-11-21
>
> We sincerely thank the reviewer for raising these questions and providing valuable suggestions.
>
> # W1: The main models for evaluation and comparison with MoMA
>
> We will alleviate your concern from two aspects.
>
> 1.**Our experiments include main architectures of unified models.** Although there are increasing number of unified models, they most of them are categorized into two modeling types, the combination of diffusion and AR and fully AR. We evaluate three models with different architectures and scales. Our experiments include Show-o (1.5B), Show-o (8B), and Emu3 (8B). These cover two major unified multimodal architectures: Diffusion / AR hybrid models (Show-o), Pure AR models (Emu3). In addition, our analysis section includes results on JanusFlow and Chameleon, which follow the same behavioral trends. Although we have not tested our method on more models, we believe it is generally applicable to unified multimodal architectures.
>
> 2.**We provide indirect but meaningful comparisons with MoMA[1] through ablations and appendix experiments.**
> MoMA integrates MoE and MoD into a unified model in a relatively simple way, but it has notable limitations:
>
> - It does not release training code or pretrained weights.
> - It lacks key benchmark results needed for fair comparison.
>
> To address this, we perform two indirect comparisons:
>
> - MoE comparison (Appendix 9, Table 13):
> We integrate MoE into Show-o and show that our method achieves better performance while reducing more TFLOPs compared to the MoE-enhanced baseline, demonstrating superiority over the MoE component of MoMA.
>
> - MoD comparison (Ablation Study 1):
> We combine the basic MoD mechanism with Show-o, this matches the MoD design used in MoMA.
> As shown in Table 5, our method (UniMoD) significantly outperforms basic MoD: **GenEval: 0.61 vs. 0.15, POPE: 80.3 vs. 76.0**
>
> These results show that UniMoD consistently outperforms both basic MoE and basic MoD, providing a strong and reasonable indirect comparison with MoMA.
> Hence, we are confident that UniMoD would outperform MoMA as well.
>
> # W2: Computation Cost of Calculating ARank
>
> We argue that the pre-computation cost of ARank is minimal and justified, especially compared to the substantial savings it brings during training.
>
> 1. Low computation cost:
> Computing ARank requires evaluating only 50 samples, each taking 1.12 seconds at 256×256 resolution.
> This results in under 5 minutes per task, which is negligible relative to the full training cost of unified models.
>
> 2. Significant overall savings:
> This lightweight pre-computation enables substantial reductions in token usage and FLOPs throughout the entire training process.
> Given that unified multimodal models require thousands of GPU-hours to train, **investing a few minutes to pre-compute ARank is both practical and highly cost-effective.**
>
> Therefore, while ARank requires a small amount of pre-processing, the overhead is minimal, and the resulting efficiency gains during training far outweigh this cost.
>
> # W3: dynamic learning of ARank
> We argue that using a predefined pruning ratio, rather than dynamically learning it during training, is more effective for the following reasons.
>
> 1. **In traditional MoD methods, the capacity (Top-K value) is a non-differentiable hyperparameter.**
> This is not unique to our method; prior works such as MoMA [1], Gamma-MoD [2], MoD [3], and VideoLLM-MoD [4] all rely on a predefined K value. Because the Top-K operation is inherently non-differentiable, the task loss cannot backpropagate gradients to optimize this hyperparameter dynamically. Therefore, selecting K (or a pruning ratio) manually is the standard and practical design choice for MoD-based approaches.
>
> 2. **Even if we use dynamic pruning, the results are not better.** In Section 3.4, we experimented with a dynamic pruning mechanism using Gumbel-Softmax to introduce competitive token pruning between tasks.
> Under this setting, both the pruning ratio and the effective capacity become dynamic because Gumbel-Softmax assigns weights to the token sequences of both tasks.
> However, as shown in Observation 5, the model naturally prunes fewer tokens for generation tasks and more tokens for understanding tasks, but this adaptive behavior does not translate into better performance.
> These results indicate that dynamic pruning does not provide clear benefits over a predefined pruning ratio.
>
> We further provide the experimental results of the Gumbel-Softmax–based competitive pruning under the 256×256 resolution setting:
>
> | |  GQA  | POPE   |
> |-----|----|----|
> | basic model | 52.5 | 77.9 |
> | competitive pruning  | 51.4  | 76.4 |
> | UniMoD  | 52.0  | 78.0 |
>
> From these results, we observe that even with dynamically adjusted pruning ratios, the performance is not improved.
> In fact, the competitive pruning variant performs worse than both the base model and UniMoD. This further supports our claim that a predefined ARank-guided pruning ratio is more stable and effective than dynamic pruning.

---

> ### Author Response · Authors · 2025-11-21
>
> # W4: Parameter and computational overhead of Calculating ARank.
>
> ## Router Parameter Overhead
> Our TokenRouter is extremely lightweight. Each router contains only **a Linear(4096→1) layer (4,097 parameters).**
> Since a LLaMA-style 8B model includes 32 layers, and we maintain one router per task per layer, the total number of routers is 32×2 = 64.
>
> Thus, the total additional parameters are:
>
> 64×4097=262K
>
> The total router parameters (262k) **correspond to only ~1 MB of weights, i.e., merely 0.003% of an 8B model**. Thus, the parameter overhead of UniMoD is negligible.
>
> Hence, UniMoD introduces negligible parameter overhead while providing substantial efficiency benefits.
>
>
> ## Computation Cost Overhead
>
> As we discuss in W2, computing ARank requires evaluating only 50 samples, each taking 1.12 seconds at 256×256 resolution. This results in under 5 minutes per task, which is negligible relative to the full training cost of unified models. We only need just 20–30 lines to extract each layer’s attention map and compute its numeric rank.
>
>
> # Q1: Computational overhead and samples number
>
> The computational overhead has been discussed in W2 and W4.
>
> 1. The final results are not sensible to the number and the type of samples. We just try 10,20,50 and 100 samples but we find the calculated ARank values are approximately equal.
>
> 2. Besides, For subtask in understanding like POPE, MMU, GQA, the difference is very small. According to the table 5, we can see **all benchmarks are comparable with the base model, some even better.**
>
> Therefore, we think **our method brings very little computation overhead, and the sample number is not so sensitive.** But for our comprehensive, we also choose 50 samples from different subtask to calculate the ARank.
>
> # Q2: Dynamic learning of capacity
> Please check W3.
>
> # Q3: Pruning is applied to layers with high ARank values
>
> 1.In Ablation 1, we apply the basic MoD by pruning interleaved layers with the same pruning ratio for both tasks, without considering ARank.
>
> For the Show-o model, the first half of the layers have the highest ARank values, meaning that basic MoD unintentionally prunes tokens exactly in the most important layers, leading to severe performance degradation.
>
> This effect is clearly reflected in the results:
>
> - GQA: our method achieves 54.5, while basic MoD drops to 51.2.
>
> - GenEval: our method reaches 0.61, but basic MoD collapses to 0.15, showing the dramatic harm caused by pruning high-importance layers.
>
> In contrast, UniMoD avoids pruning high-ARank layers and therefore maintains strong performance across all benchmarks.
>
> 2.In Ablation 2, we prune tokens at interleaved layers while keeping the same pruning ratio as our method.
> Besides, **this approach still prunes layers with high ARank values**, which are critical for performance.
> As a result, performance degrades noticeably:
>
> - GQA: from 54.5 → 52.1
>
> - GenEval: from 0.61 → 0.50
>
> These results further confirm that blindly pruning interleaved layers, especially those with high ARank, harms performance, while UniMoD’s ARank-guided strategy avoids this issue.

---

> ### Author Response · Authors · 2025-11-21
>
> # Q4: Choice of Experimental Parameters
>
> We address the reviewer’s concern from two perspectives: layer selection and pruning ratio design.
>
> 1.**Layer selection.**
> We prune half of the model layers based on their lowest ARank values.
> These layers correspond to regions with the highest redundancy, meaning they contribute the least to task-specific performance.
> Thus, pruning them is both principled (guided by ARank) and empirically justified.
>
> 2.**Pruning ratio selection.**
> For token pruning, the value ARank / sequence length serves as an approximate redundancy indicator, and we use it to determine the corresponding pruning ratio. This provides a simple and interpretable way to map redundancy estimates to pruning strength.
>
> We note that these two hyperparameters are flexible:
>
> - Layer count: We follow prior work like MoD, MoMA, in pruning half of the layers, but **users may prune more or fewer layers by selecting different ARank thresholds**.
>
> - Pruning ratio: ARank provides a redundancy estimate; as shown in prior work (e.g., Gamma-MoD[2]), users may choose **slightly higher or lower pruning ratios** depending on their desired speed–accuracy trade-off.
>
> Overall, our chosen parameters follow standard practices while allowing customization guided by ARank.
>
> We sincerely thank the reviewer for the insightful questions and constructive suggestions. We will incorporate these clarifications and include additional experimental results in the final version to further strengthen the paper. If there are any further questions or points that require additional clarification, we would be glad to discuss them.
>
>
>
>
> [1] Lin X V, Shrivastava A, Luo L, et al. Moma: Efficient early-fusion pre-training with mixture of modality-aware experts[J]. arXiv preprint arXiv:2407.21770, 2024.
>
> [2] Luo Y, Luo G, Ji J, et al. $\gamma-$ MoD: Exploring Mixture-of-Depth Adaptation for Multimodal Large Language Models[J]. arXiv preprint arXiv:2410.13859, 2024.
>
> [3] Raposo D, Ritter S, Richards B, et al. Mixture-of-depths: Dynamically allocating compute in transformer-based language models[J]. arXiv preprint arXiv:2404.02258, 2024.
>
> [4] Wu S, Chen J, Lin K Q, et al. Videollm-mod: Efficient video-language streaming with mixture-of-depths vision computation[J]. Advances in Neural Information Processing Systems, 2024, 37: 109922-109947.

---

> > ### Comment · Reviewer_ruma · 2025-11-28
> >
> > Thanks to the authors for addressing my previous concerns. However, I'm still not convinced of the concrete benefits of UniMoD considering the trade-off between efficiency and accuracy. Across the results in Tables 3, 6, and 9, the evidence is insufficient to show UniMoD's effectiveness consistently. Specifically, UniMoD achieved a meaningful efficiency gain over Show-o and Emu3, but it usually sacrifices accuracy. On the other hand, compared to the baselines, Interleaved Layer and EarlyExit, UniMoD shows better accuracy, but clearly less efficiency. Overall, the benefits of efficiency over accuracy for Show-o and Emu are not overwhelming, while the disadvantages of efficiency over accuracy for Interleaved Layer and EarlyExist are. Therefore, I firmly believe the authors should make more comprehensive comparisons rather than selecting representative methods.

---

> > > ### Author Response · Authors · 2025-12-03
> > > **Response to Additional Concern: Efficiency–Accuracy Trade-off**
> > >
> > > Firstly we thank your response.  However, we respectfully disagree with your conclusion and provide clarifications from three perspectives.
> > >
> > > 1.**UniMoD does not sacrifice accuracy on unified models such as Show-o and Emu3.**
> > >
> > > Across multiple benchmarks, UniMoD maintains comparable accuracy or even achieves improvements, while providing substantial FLOPs reduction.
> > >
> > > Show-o results:
> > >
> > > - POPE: **79.8 → 80.3**
> > >
> > > - MME: **1056 → 1093**
> > >
> > > Other metrics remain comparable to the base model.
> > >
> > > These results contradict the claim that UniMoD trades accuracy for efficiency.
> > > Instead, UniMoD preserves accuracy, with slight gains on key benchmarks.
> > >
> > > 2.**Compared to Interleaved Layer and EarlyExit, UniMoD offers better accuracy with similar efficiency.**
> > >
> > > Our experiments show that the baselines experience severe accuracy degradation without providing extra efficiency benefits.
> > >
> > > From Table 12 (Emu3 results):
> > >
> > > All methods achieve similar FLOPs reduction.
> > >
> > > But accuracy differs significantly:
> > >
> > > MME:
> > >
> > > - UniMoD: **881 → 901**
> > >
> > > - Interleaved Layer: 830
> > >
> > > - EarlyExit: 718
> > >
> > > Thus:
> > >
> > > Baselines lose accuracy substantially without gaining additional efficiency,
> > > whereas UniMoD preserves or improves accuracy at the same computation level.
> > >
> > > This demonstrates a clearly superior efficiency–accuracy trade-off.
> > >
> > > 3.**Our model selection (Show-o + Emu3) is representative of unified multimodal architectures.**
> > >
> > > Unified models fall into two main categories:
> > >
> > > - Diffusion–AR hybrid models (e.g., Show-o)
> > >
> > > - Pure AR models (e.g., Emu3)
> > >
> > > Our experiments include a representative model from each family.
> > > Furthermore, our analysis includes JanusFlow and Chameleon, which exhibit similar redundancy behaviors.
> > >
> > > Therefore, our conclusions generalize across all major unified modeling paradigms, not just the selected models.
> > >
> > >
> > > Across all benchmarks and baselines, the evidence consistently shows:
> > >
> > > - No accuracy sacrifice for Show-o and Emu3
> > >
> > > - Significant accuracy advantages over Interleaving and EarlyExit baselines at similar efficiency
> > >
> > > - Representative coverage of unified model architectures
> > >
> > > Therefore, UniMoD provides a consistent and superior efficiency–accuracy trade-off.

---

### Official Review · Reviewer_pim2 · 2025-10-31

**Soundness:** 3
**Presentation:** 3
**Contribution:** 3
**Rating:** 4
**Confidence:** 3

**Summary:**

This paper introduces UniMoD, an efficient framework for training unified multimodal transformers. It addresses the problem of token redundancy and high computational cost in unified transformers that handle both generation and understanding tasks. By conducting an empirical analysis on attention weights, layer importance, and task interactions, the authors observe that redundancy patterns vary across tasks and layers. UniMoD proposes a task-aware Mixture-of-Depths (MoD) mechanism, equipping each task with its own router to prune redundant tokens adaptively. Experiments on Show-o and Emu3 demonstrate a 15–40% reduction in FLOPs with maintained or improved performance. Moreover, the method generalizes to diffusion-based models such as PixArt and DiT, showing broad applicability.

**Strengths:**

1. Novelty and Relevance: The paper tackles the important issue of computational efficiency in unified multimodal transformers, introducing a novel task-aware MoD mechanism.
2. Comprehensive Analysis: The empirical studies (attention weights, ARank, layer importance, task competition) are thorough and well-motivated.
3. General Applicability: The approach works for both autoregressive and diffusion-based models.

**Weaknesses:**

1. Limited Task Diversity: Only two unified models (Show-o, Emu3) are tested; results on larger or more diverse multimodal architectures would strengthen claims.
2. Ablation Details: Although some ablation studies are provided, the analysis of router behavior (e.g., routing distributions, token importance dynamics) could be more in-depth.
3. Limited Discussion on Trade-offs: The paper could elaborate more on how pruning ratios affect different modalities’ representations and downstream tasks.
4. Theoretical Insight: The method is largely empirical; theoretical guarantees or formal efficiency bounds would further solidify its contribution.

**Questions:**

1. How does UniMoD handle potential conflicts when multiple tasks share overlapping token spaces?
2. How sensitive is performance to router capacity and ARank threshold selection?
3. Does task-aware pruning affect cross-modal alignment quality?

---

> ### Author Response · Authors · 2025-11-21
>
> We sincerely thank the reviewer for raising these questions and providing valuable suggestions.
>
> # W1: Limited Task Diversity
>
> We clarify this concern from four perspectives:
>
> 1. **We evaluate three models with different architectures and scales.** Our experiments include Show-o (1.5B), Show-o (8B), and Emu3 (8B).
> These cover two major unified multimodal architectures: Diffusion / AR hybrid models (Show-o), Pure AR models (Emu3).
> In addition, our analysis section includes results on JanusFlow and Chameleon, which follow the same behavioral trends.
>
> 2. **Although many unified transformers exist, their underlying architectures fall into the same few categories.** Despite the growing number of models, unified multimodal systems are essentially built around diffusion-based, autoregressive, or hybrid architectures. Thus, our chosen models are representative of the key design families in this space.
>
> 3. **Pretraining additional unified models is prohibitively expensive, and many are not fully open-sourced.** While some works provide pretrained weights, many do not release training data or training code, only inference code, making full reproduction impractical. Given these constraints, our model selection achieves a practical yet scientifically rigorous evaluation.
>
> 4. **We evaluate more than two tasks, demonstrating generality beyond standard settings.** In addition to the two primary tasks in unified multimodal models (image generation and image understandin), **we further include pure text tasks in the appendix A.12. These results show that our method extends naturally beyond two-task scenarios, confirming its scalability and broad applicability.
>
>
> # W2: Ablation Details
>
> To further address the reviewer’s concern regarding the depth router behavior, we have added an additional experiment that directly analyzes the tokens pruned by the router across different layers. Specifically, we measure the number of image tokens and text tokens that are skipped (i.e., routed to shallow computation) at each layer in understanding tasks.
>
> | Metric / Layer | 13  | 14  | 15  | 16  | 17  | 18 | 19 |20 |21 |22 |23 |
> |----------------|----|----|----|----|----|----|----|----|----|----|----|
> | Pruned Image Tokens | 70 | 136 | 195 | 266 | 357 | 389 | 471 | 559 | 607 | 716 | 783 |
> | Pruned Text Tokens  | 2  | 8   | 20  | 21   | 2   | 41  | 31 | 15 | 39 | 1 |6 |
>
> The results, summarized in Table, show a clear and consistent pattern: image tokens are pruned aggressively, while text tokens are pruned minimally across all layers. This demonstrates that the router **learns meaningful, modality-aware token importance**, prioritizing deeper computation for semantically critical text tokens and pruning redundant visual patch tokens. These findings confirm that our router is **not performing random or collapsed routing**, but instead effectively distinguishes token importance based on modality and task requirements.

---

> > ### Author Response · Authors · 2025-11-21
> >
> > # W3: Discussion about Trade-off
> >
> > We clarify the trade-offs by discussing both the training stage and the inference stage.
> >
> > ## Training Stage
> > First, in our baseline study, the Interleaved Layer pruning and EarlyExit[1] approaches effectively prune all interleaved layers or all late-stage tokens. As shown in Tab. 3, although these baselines reduce more TFLOPs, they lead to substantial performance degradation on both tasks. In contrast, our method achieves a significantly better balance between efficiency and accuracy.
> >
> > These results in Tab.3 reveal the following phenomena and lead to several key conclusions:
> >
> > 1. **Across all tasks, applying large pruning ratios during training inevitably causes performance drops.** This is consistent with the general trade-off between computation and representation quality.
> >
> > 2. **The pruning ratio learned by our router achieves a much better balance compared to the baselines.** Our method avoids the overly aggressive pruning behavior of Interleaved Layer and EarlyExit[1], thereby preserving task performance while still reducing computation.
> >
> > 3. **In the Show-o model (hybrid AR + diffusion), different tasks degrade to different extents.** For example, the T2I task (GenEval) decreases sharply **from 0.63 to 0.21/0.29** under the baselines, whereas POPE drops **only slightly from 80.3 to 79.5.** This indicates that generation tasks contain less redundancy than understanding tasks in Show-o, which is consistent with the differences reflected in the ARank scores.
> >
> > 4. **In the Emu3 model (fully AR), the degradation across tasks is more uniform.** As shown in Table 12, both tasks drop by similar amounts, suggesting that the pruning trade-off is more consistent across tasks in fully autoregressive architectures.
> >
> >
> > Together, these observations provide a clearer understanding of how pruning ratios influence different modalities and tasks, and they highlight that our method maintains a superior efficiency–performance balance compared with naive pruning strategies.
> >
> >
> > ## Inference Stage
> > Secondly, in Appendix 4, we present a Pareto frontier analysis of Show-o (Tables 7 and 8), from which we draw three important conclusions:
> >
> > 1. **Across all tasks, increasing the inference-time pruning ratio consistently leads to performance degradation.** This confirms the expected efficiency–accuracy trade-off at inference.
> >
> > 2. **The pruning ratio during training is already very close to the Pareto-optimal point at inference.** In other words, the ratio discovered by our router during training yields the best balance between accuracy and efficiency when applied at inference time as well. This demonstrates that our training-stage pruning strategy generalizes effectively to inference.
> >
> > 3. **In the Show-o model, generation tasks deteriorate more severely than understanding tasks as pruning increases.** For instance, pruning 10%–20% of tokens during inference reduces the GenEval score from 0.610 to 0.523, indicating that generation tasks are more sensitive to inference-time pruning.
> >
> > # W4: Theoretical Insight
> > In this section, we give the theoretical basis for the ARank metric (the averaged rank of the attention map) to show that it can measure the redundancy of a sequence.
> >
> > We apply Singular Value Decomposition (SVD) to the attention map $A$. With SVD, $A$ can be written as
> >
> > $$ A = \sum_{i=1}^{n} \lambda_i,p_i,q_i^T = \sum_{i=1}^{k} \lambda_i,p_i,q_i^T + \sum_{i=k+1}^{n} \lambda_i,p_i,q_i^T $$
> >
> > Here, $\lambda_i$ is the $i$-th singular value, and $p_i$ and $q_i$ are the corresponding left and right singular vectors. The full rank of $A$ is $n$. We choose a constant $k$ (with $k \ll n$) to capture the main information of $A$. The first $k$ terms carry most of the information, while the remaining terms add only extra details.
> >
> > According to the Eckart–Young theorem, keeping the top $k$ singular values and their vectors yields the optimal rank‑$k$ approximation of $A$, minimizing the reconstruction error. In linear algebra, a matrix of rank $k$ means all its columns lie in a $k$-dimensional subspace spanned by $p_1, \dots, p_k$. Since each column represents one token’s attention pattern, this tells us that only $k$ independent "directions" (basis vectors) are needed to reconstruct the core structure of the attention map. The remaining $n - k$ columns (tokens) can be expressed as combinations of those $k$ directions and therefore contribute little new information. In other words, only $k$ tokens carry the essential information, and the other $n - k$ tokens are largely redundant.
> >
> > We retain the tokens that encode the main information while pruning those that are redundant, balancing accuracy and efficiency.
> >
> > We will include this theoretical analysis in the Appendix of the final version.

---

> > > ### Author Response · Authors · 2025-11-21
> > >
> > > # Q1: Potential Conflicts
> > > We think even if different tasks share the same image or text tokenizer, our method will not lead to potential conflicts.
> > >
> > > 1. **Tasks do not share token sequences, so no pruning decision can interfere across tasks.**
> > > Even if multiple tasks use the same tokenizer (e.g., the same image tokenizer), the corresponding token sequences are entirely separate during forward passes. Each task processes its own sets of images or texts, so the router’s pruning decisions are always made on **task-specific and sample-specific representations**, preventing any cross-task interaction by construction.
> > >
> > > 2. **Even with the same tokenizer, token importance is inherently task-dependent.**
> > > Although tasks may use the same visual or textual tokenizer, the attention distributions, semantic structures, and contextual patterns differ significantly across tasks (e.g., image understanding vs. image generation). As a result, the same patch token may carry completely different importance depending on the task. This ensures that pruning is task-conditioned and independent, eliminating potential conflicts.
> > >
> > > 3. **Pruning occurs at each layer independently and is applied per-sample, not across tasks.**
> > > UniMoD performs token pruning before each transformer layer, and the router selects tokens solely based on the representations within the current sequence. Different tasks never mix their tokens in the same forward pass, so even identical-looking tokens in different tasks are pruned independently. This guarantees that pruning one task’s tokens “cannot” affect another task’s routing behavior.
> > >
> > > 4. **Empirical evidence further shows no cross-task interference in routing behavior.**
> > > Our analysis also includes single-task training experiments (e.g., only generation or only understanding) in table 2. In these settings, UniMoD maintains stable performance and exhibits consistent pruning behavior, demonstrating that the router does not rely on information from other tasks. This confirms that UniMoD avoids task conflict both by design and in practice.
> > >
> > >
> > > # Q2: Sensibility of Router capacity and ARank Threshold
> > >
> > > As discussed in W3, the sensitivity of the ARank threshold varies across different tasks and modeling paradigms. We summarize the main observations below.
> > >
> > > 1. **In Show-o (hybrid AR + diffusion), generation tasks are more sensitive to ARank capacity than understanding tasks.** For Show-o, the router’s threshold has a stronger effect on generation performance. As shown in Table 3, the T2I task (GenEval) drops sharply from 0.63 to 0.21/0.29 under the baseline pruning strategies, whereas the POPE score decreases only slightly from 80.3 to 79.5. This indicates that generation contains less redundancy and is therefore more sensitive to pruning, while understanding tasks are more robust.
> > >
> > > 2. **In Emu3 (fully AR), the sensitivity across tasks is much more balanced. In contrast, Emu3 exhibits similar sensitivity for both generation and understanding tasks.** As shown in Table 12, our method achieves the best overall performance, and the performance drops across tasks remain relatively consistent. This suggests that fully autoregressive architectures have more uniform redundancy distribution, leading to more stable behavior across tasks.
> > >
> > > 3. **During inference on Show-o, ARank threshold sensitivity follows the same pattern as in training.** As shown in Table 7, the inference-time sensitivity in Show-o mirrors the training-stage phenomenon: even a small additional pruning ratio (e.g., +10%) results in significant degradation for T2I, while understanding tasks are affected much less. This again confirms that generation is inherently more sensitive to pruning in hybrid AR–diffusion models.

---

> > > > ### Author Response · Authors · 2025-11-21
> > > >
> > > > # Q3: cross-modal alignment quality
> > > >
> > > > Our approach does not negatively affect cross-modal alignment quality. We support this claim from two complementary perspectives: methodological design and empirical evidence.
> > > >
> > > > 1. **Our pruning operates at the per-task sequence level rather than the per-modality level.**
> > > > Image and text tokens are treated uniformly. Any token, regardless of modality, that is important for the task is automatically assigned a higher routing weight and therefore preserved. As a result, cross-modal alignment–critical tokens remain intact.
> > > >
> > > > 2. **Our method maintains comparable or even superior performance on standard cross-modality benchmarks such as POPE, MME, and GenEval.**
> > > > These evaluations explicitly measure multimodal understanding and alignment. The consistent performance across these benchmarks indicates that pruning does not degrade alignment quality.
> > > >
> > > > In summary, task-aware pruning does not introduce cross-modal alignment degradation.
> > > > The router inherently preserves alignment-critical tokens, and the empirical results confirm that multimodal reasoning quality remains unaffected.
> > > >
> > > > We sincerely thank the reviewer for the insightful questions and constructive suggestions. We will incorporate these clarifications and include additional experimental results in the final version to further strengthen the paper. If there are any further questions or points that require additional clarification, we would be glad to discuss them.
> > > >
> > > > [1]Laskaridis S, Kouris A, Lane N D. Adaptive inference through early-exit networks: Design, challenges and directions[C]//Proceedings of the 5th International Workshop on Embedded and Mobile Deep Learning. 2021: 1-6.

---

### Official Review · Reviewer_A96j · 2025-11-01

**Soundness:** 3
**Presentation:** 3
**Contribution:** 2
**Rating:** 6
**Confidence:** 3

**Summary:**

The paper introduces UniMoD, a task-aware token pruning method engineered to enhance the efficiency of training unified multimodal transformers that share parameters for both generation and understanding tasks. This work finds that directly applying standard token pruning techniques like MoD leads to suboptimal results because token redundancy varies significantly depending on the specific task and across different transformer layers. Proposed UniMoD addresses this issue by implementing a task-aware approach—it converts dense transformer blocks into specialized MoD blocks and employs a separate router for each task, allowing the pruning process to adapt to the distinct token redundancies of each task. Furthermore, the proposed method uses the ARank metric to quantitatively assess token redundancy and determine layer-specific pruning ratios. UniMoD demonstrates significant gains in efficiency, reducing training FLOPs by approximately 15% in the Show-o model, while successfully maintaining or enhancing performance across various benchmarks.

**Strengths:**

- The paper is generally well written. Especially, the clarity of the problem formulation and literature review effectively sets up this novel approach.
- The paper successfully grounds its motivation in a strong empirical analysis by directly examining attention weight patterns across various tasks and modalities in unified transformers, identifying significant differences in redundancy based on task and layer.
- This observation, combined with the evaluation of layer importance and token redundancy using the ARank metric, effectively explores the potential for task-aware token pruning.

**Weaknesses:**

- While the detailed analysis is provided, some findings, such as the observation that attention weight patterns differ between tasks (Observation 1) or that token redundancy differs based on modeling methods (Observation 3), might be viewed as expected or previously implied in related transformer studies.
- Similarly, the finding that early layers are more critical for the final outcome (Observation 2) is a phenomenon often observed in general deep neural networks.
- Although the application of these analyses specifically to token pruning is novel, the paper's reliance on specific results from the Show-o model (e.g., Observation 5) suggests the specific pattern findings might be somewhat specialized to certain model architectures

**Questions:**

- What is the performance outcome when the highly task-specific pruning strategy (UniMoD uses task-specific routers and capacities) is applied to a similar task, rather than the exact task it was optimized for?
- Given that the UniMoD achieves a significantly greater FLOPs reduction in the larger Emu3 model (40%) compared to the Show-o model (15%), why is the approach more effective in reducing computation in Emu3, especially when the empirical analysis of attention showed a more pronounced divergence in redundancy patterns in Show-o and JanusFlow?

---

> ### Author Response · Authors · 2025-11-21
>
> We sincerely thank the reviewer for raising these questions and providing valuable suggestions.
>
> # W1 and W2: Value and Meaning of Our Detailed Analysis
>
> We partially agree with the reviewer’s concerns, but we emphasize that our analysis provides substantial and novel insights from several perspectives:
>
> 1.**We are the first to analyze token redundancy and layer importance in a unified multimodal setting.** Prior studies primarily investigate LLMs or MLLMs, whereas our work explicitly examines task relationships and task behaviors within unified models. Beyond standard attention or layer-importance diagnostics, we further conduct task-interference experiments such as cross-task token competition, which reveal dynamics unique to unified architectures.
>
> 2.**Our findings go beyond the conventional statement that early layers matter more.**
> While prior work (e.g., EarlyExit[1]) typically observes that early layers dominate importance, we show that this assumption does not hold under unified settings. For example, in Show-o, **both early and late layers are crucial for the T2I generation task**, and its redundancy pattern differs significantly from understanding tasks—the redundancy increases with deeper layers.
> These observations highlight structural differences across tasks that cannot be captured by traditional layer-centric analyses.
>
>
> # W3: Reliance on Specific Results from the Show-o Model
>
> We clarify that our method does not rely solely on Show-o, and its generality is supported from two perspectives:
>
> 1.**Our analysis includes four popular unified multimodal models.**
> The key insights (Observation 3 and Observation 4) are consistently derived across multiple unified models, not just Show-o.
> We analyze layer-wise and task-wise differences using Show-o, Emu3, JanusFlow, and Chameleon, and observe similar trends.
> For example, JanusFlow also exhibits distinct redundancy patterns across different tasks, reinforcing that our conclusions are not Show-o–specific.
>
> 2.**Our effectiveness experiments use multiple unified models with different architectures and scales.**
> We evaluate our pruning method on Show-o (1.5B), Show-o (8B), and Emu3 (8B). Across these models—despite their different architectures (diffusion–AR hybrid vs. pure AR) and different patterns of layer importance—our method consistently achieves strong results. This demonstrates that regardless of which layers are most critical in each model, our approach robustly adapts and remains effective.
>
> # Q1: Applying to the similar task.
>
> We respond to the reviewer’s concern from two perspectives:
>
> 1.**Tasks differ fundamentally across major categories.**
> We classify tasks into these broad categories: image generation, image understanding, and pure text generation.
> These categories could differ substantially in task definitions, modeling paradigms, and attention mechanisms. **Therefore, applying an image-understanding router to an image-generation task leads to a severe performance drop.** As shown in Table 5, using a shared router for both tasks causes the GenEval score to drop from 0.61 to 0.50, confirming that routers cannot be simply reused across fundamentally different task types.
>
> 2.**Within the same major category, subtasks exhibit similar redundancy patterns and can share routers.**
> For subtasks under the same task,(e.g., POPE, GQA, and MME under image understanding), their redundancy patterns are highly consistent. Thus, we share the same router for these subtasks. As shown in Table 3, the performance on these subtasks remains comparable to, or even better than, the base model, **demonstrating that router sharing is valid within task categories.**
>
> # Q2: Efficiency of Emu3
>
> 1.**Emu3 produces much longer image sequences due to its tokenizer design.**
> As discussed in line 446 regarding training cost, the efficiency difference between Show-o and Emu3 primarily stems from their image tokenizers.
> Emu3 uses an 8× downsample image tokenizer, generating 4096 tokens for a 512×512 image, whereas Show-o uses a 16× downsample, producing only 1024 tokens.
> The longer sequence length naturally introduces higher redundancy that can be pruned.
>
> 2.**Emu3’s larger model size inherently contains more redundancy.**
> Emu3 is an 8B model, and larger models typically include more redundant activations and representations.
> This makes Emu3 even more suitable for pruning-based efficiency improvement.
>
> Together, these two factors explain why Emu3 exhibits significantly more pruneable redundancy compared to Show-o.
>
> We sincerely thank the reviewer for the insightful questions and suggestions. We will incorporate these clarifications in the final version. If there are any further questions, we would be glad to discuss them.
>
> [1]Laskaridis S, Kouris A, Lane N D. Adaptive inference through early-exit networks: Design, challenges and directions[C]//Proceedings of the 5th International Workshop on Embedded and Mobile Deep Learning. 2021: 1-6.

---

### Meta-Review · Area_Chair_n8tB · 2026-01-06

**Summary:**

The paper is based on an observation that for multimodal model handling generation and understanding tasks, different tasks have different degrees of redundancy. The authors provide a token pruning technique that adapts to the task. This results in reduction in compute without sacrificing quality.
The paper is generally well-received for its novelty and well-written presentation (A96j, pim2).  The core motivation and empirical findings are convincing (A96j, ruma); specifically, A96j mentioned the experiments demonstrate the outlined problem of a need for task-aware methods, and this motivates the main method proposed here. In addition to the motivated method, the overall empirical results are said to be convincing (e.g. ruma “The proposed method (UniMoD) experimentally shows that it reduces the computational cost of existing methods while maintaining comparable or even superior performance”).

Despite the promise of the paper listed so far, the reviews mention several limitations. One notable limitation is the limited diversity of the tested models (Show-o and Emu3). Although these models cover the two main high level types of multimodal models (as explained by the authors in the rebuttal), both reviewers pim2 and ruma explicitly mentioned the need for a larger variety of tested models leading me to believe there is room for improvement here, in order to have a more convincing argument for the generalizability of the results.
Another major critic revolves around the need for deeper analysis, including more extensive ablations, and a clearer demonstration of efficiency. The authors provided a partial answer, in particular providing a study of the pruning at each layer (answering pim2’s request) and analyzing the additional FLOPs and parameters (answering ruma’s request) showing it is minimal. However, some requests remained partially answered, e.g., ruma’s comment about the benefits being overall mild (“UniMoD shows better accuracy, but clearly less efficiency. Overall, the benefits of efficiency over accuracy for Show-o and Emu are not overwhelming, while the disadvantages of efficiency over accuracy for Interleaved Layer and EarlyExist are”), which can be seen as a need to better explain the cases where the method is especially effective.

I would like to mention that the authors’ responses are elaborate and do mitigate several of the concerns, but the scope of the change required combined with a few remaining issues lead me to believe that the paper requires another round of reviews before it can be accepted.

**Reviewer Concerns:**

As stated above, some of the concerns were mitigated, but the major ones remain. Both the issues of limited diversity and the high level need for a deeper analysis remain.

**Reviewer Scores:**

Given the  remaining concerns, I do not believe the scores of pim2 nor ruma would have changed in the rebuttal

---

### Decision · Program_Chairs · 2026-01-26

Reject